



# Current estimates of biogenic emissions from Eucalypts uncertain for Southeast Australia

K.M. Emmerson[1], I.E. Galbally[1], A.B. Guenther[2], C. Paton-Walsh[3], E-A. Guerette[3], M.E. Cope[1], M.D. Keywood[1], S.J. Lawson[1], S.B. Molloy[1], E. Dunne[1], M. Thatcher[1], T. Karl[4], S.D. Maleknia[5]

5   1 CSIRO Oceans & Atmosphere, PMB1, Aspendale, VIC, Australia.
2 Department of Earth System Science, University of California, Irvine, USA.
3 Centre for Atmospheric Chemistry, School of Chemistry, University of Wollongong, Wollongong, NSW, Australia.
Institute of Atmospheric and Cryospheric Sciences, University of Innsbruck, Innsbruck, Austria.
Centre for Ecosystem Science, School of Biological, Earth and Environmental Sciences, University of New South Wales,
Australia.

*Correspondence to*: K.M. Emmerson (kathryn.emmerson@csiro.au)

**Abstract.** The biogenic emissions of isoprene and monoterpenes are one of the main drivers of atmospheric photochemistry, including oxidant and secondary organic aerosol production. In this paper, the emission rates of isoprene and monoterpenes
from Australian vegetation are investigated for the first time using the Model of Emissions of Gases and Aerosols from Nature version 2.1 (MEGANv2.1), the CSIRO chemical transport model, and atmospheric observations of isoprene, monoterpenes and isoprene oxidation products (methacrolein and methyl-vinyl-ketone). Observations from four field campaigns during three different seasons are used, covering urban, coastal suburban and inland forest areas. The observed concentrations of isoprene and monoterpenes were of a broadly similar magnitude, which may indicate that southeast Australia holds an unusual position
where neither chemical species dominates. The model results overestimate the observed atmospheric concentrations of isoprene (up to a factor of 6) and underestimate the monoterpene concentrations (up to a factor of 4). This may occur because the emission rates currently used in MEGANv2.1 for Australia are drawn mainly from young Eucalypt trees (<7yrs), which may emit more isoprene than adult trees. There is no single increase/decrease factor for the emissions which suits all seasons and conditions studied. There is a need for further field measurements of in-situ isoprene and monoterpene emission fluxes in
Australia.

## 1   Introduction

Biogenic volatile organic compounds (BVOCs) originate from terrestrial and marine ecosystems, and have an annual flux in the region of 1150 Tg C yr$^{-1}$ (Guenther et al., 1995). Approximately 90% of BVOCs are emitted from plants and trees, with
the most dominant species being isoprene and monoterpenes (Lathiere et al., 2006; Guenther et al., 2012). The isoprene and monoterpene emission rates from vegetation are determined by a combination of environmental factors (light, temperature, water stress etc.) and genetic make-up of the species being considered (Guenther et al., 2012). In regions of dense vegetation these BVOCs dominate the oxidative capacity of the atmosphere (Houweling et al., 1998; Taraborrelli et al., 2012), and are important in the production of ozone (Simpson, 1995; Pierce et al., 1998) and secondary organic aerosol (Hoffmann et al.,
1997; Griffin et al., 1999; van Donkelaar et al., 2007).

Concentrations of BVOCs in the atmosphere are a function of the emission rate from the underlying vegetation, the mixing depth of the boundary layer, entrainment rate at the top of the boundary layer, horizontal advection, and the rate of removal



within the boundary layer by the hydroxyl and nitrate radicals, and ozone. All these processes vary diurnally. Modern chemical transport models can simulate all these processes provided they include an emission module for BVOCs such as the Model of Emissions of Gases and Aerosols from Nature (MEGAN).

MEGAN was developed to provide a parameterisation for BVOC emissions applicable over the Earth's surface (Guenther et al., 2012). MEGAN uses meteorological parameters such as temperature and solar radiation, land use maps incorporating vegetation and land cover, and emission factors based on global observations of plant responses to light and temperature. MEGAN has been incorporated and run within a number of global chemistry models (Guenther et al., 2006; Heald et al., 2008; Emmons et al., 2010; Millet et al., 2010; Pfister et al., 2008), and for regional air quality studies (Situ et al., 2013; Stavrakou et al., 2014; Kim et al., 2014). Sensitivity studies on the input data for MEGAN have highlighted the importance of time and spatial resolution in meteorological data (Ashworth et al., 2010; Arneth et al., 2011). A comparison of isoprene emissions driven by low resolution (degree scale) and high resolution (10km) meteorological fields showed changes up to 150% due to smoothing via averaging effects (Pugh et al., 2013). The importance of using accurate land cover data in respect to the effects of isoprene on ozone concentrations has also been discussed (Kim et al., 2014), as has changing all vegetation from default species to Eucalypts (Situ et al., 2013), which increased isoprene concentrations by 315%.

There are over 700 species of Eucalypts native to Australia, many of which are expected to contribute to isoprene emissions in the Southeast region. Evans et al. (1982) reported the first comprehensive survey of isoprene emission and found that *Eucalyptus globulus* was the highest isoprene emitter of the 54 plant species examined. Eucalypts were selected to be the subject of a number of subsequent isoprene emission studies and are considered as among the highest isoprene emitting plants (e.g., Loreto and Delfine 2000). A small number of BVOC emission measurements have been made in Australia, particularly of Eucalypt species (Winters et al., 2009; He et al., 2000), flowering plants and pasture, including grass cutting (Kirstine et al., 1998) and tropical grasslands/woodlands (Ayers and Gillett, 1988). Emissions from Eucalypt species outside Australia have been measured in the field (Street et al., 1997), and the laboratory (Guenther et al., 1991).

Previous MEGAN predictions of BVOC emissions across Australia have had limited success. Muller et al. (2008) found an overestimate of isoprene across northern Australia, and in subsequent work used satellite measurements of formaldehyde to suggest the overestimate is a factor of 2-3 in January (Stavrakou et al., 2015). Sindelarova et al. (2014) found that reductions of 50% in Australian isoprene emissions could be achieved by accounting for reduced isoprene emissions during low soil moisture conditions.

The imperative for understanding biogenic emissions from Australia is clear as the country covers 22% of the land area in the Southern Hemisphere (excluding Antarctica). This is the first high resolution regional study focussing on whether the emission factors used in MEGAN are appropriate to represent BVOC emissions from diverse locations in southeast Australia. We compare atmospheric concentrations of isoprene and monoterpenes observed in these locations with concentrations modelled using MEGAN, the default emission factors and the CSIRO chemical transport model. Sensitivity studies are undertaken on these emission factors. Tests on other variables to assess model uncertainty are shown in the supplementary material. Differences between the modelled and measured BVOCs are critically examined and the need for revised regional emission factors are evaluated.


## 2    Materials and Methods

### 2.1    Field experiments

Gas phase biogenic VOC data were measured using a Proton Transfer Reaction Mass Spectrometer (PTR-MS) collected during four field experiments in areas of diverse land cover in southeast Australia. Figure 1 shows a map giving the locations of the

field campaign sites in southeast Australia, showing their proximity to the coast and urban regions, and forested areas. Data within Figure 1 are discussed later. The PTR-MS measures groups of species which correspond to certain mass to charge (m/z) ratios, for example isoprene, $C_5H_8$, is identified at m/z = 69 (made up of the mass of $C_5H_8$ (68 g mol$^{-1}$) and a proton (1 g mol$^{-1}$)). Whilst monoterpenes are identified at both m/z = 137 and 81 (a dominant fragment produced by dissociative proton transfer), only the m/z=137 will be used. The PTR-MS technique is ideal for developing and evaluating parameterisations for

lumped species modelling as most chemical mechanisms do not separate individual monoterpenes such as α- and β- pinenes, and conventional gas chromatographic techniques may underestimate the actual monoterpene loading (Lee et al., 2005). Hourly averages have been calculated from the PTR-MS data to be comparable to the time period of the modelled output. For details of the PTR-MS measurements please refer to the citations given for each field campaign.

### 2.1.1    The Sydney Particle Study

The Sydney Particle Study (SPS) took place at Westmead, 26km to the west of Sydney centre (150.9961°E, 33.8014°S) (Cope et al., 2014). The site is situated in a grassy field within the grounds of a psychiatric hospital. Two intensive field campaigns took place; SPS1 which occurred between February 1$^{st}$ and March 7$^{th}$ 2011 (summer) and SPS2 between April 14$^{th}$ and May 14$^{th}$ of 2012 (autumn). The PTR-MS was operational from February 18th during SPS1, and throughout the whole of SPS2. The height of the inlet was approximately 4m.

### 2.1.2    MUMBA

The Measurement of Urban, Marine and Biogenic Air (MUMBA) field campaign took place between December 21$^{st}$ 2012 and February 16$^{th}$ 2013 (summer) at the University of Wollongong eastern campus (150.8995°N, 34.3972°S), about 80km to the south of Sydney (Paton-Walsh et al., submitted). Wollongong is a coastal location with sharp gradients between marine, urban and forested regions. The PTR-MS instrument was situated in a hut surrounded by a grass field, and sampled from a mast at a

height of ~10m above the surrounding ground level.

### 2.1.3    Tumbarumba

PTR-MS measurements were made for one week at Tumbarumba in New South Wales (148.1517°E, 35.6566°S) between November 8$^{th}$ – 14$^{th}$ 2006 (late spring) (Maleknia, 2012; Maleknia et al., 2009). Tumbarumba is located within the Bago State Forest and is surrounded by dominant Eucalypt species of *E. delagatensis* (Alpine Ash) and *E. dalrympleana* (Mountain Gum)

with an average height of 40m. Isoprene and monoterpenes were observed from an inlet height of 45m. Despite being late spring the campaign experienced snowstorm conditions that caused damage to the trees. This resulted in a four-fold increase to the emission pattern of monoterpenes whilst isoprene levels remained low due to cold temperatures (~8°C) (Maleknia et al., 2009). Three days of eddy covariance flux measurements are available for isoprene and monoterpenes from the post-storm period at Tumbarumba. These data will provide a direct constraint on modelled emissions despite being caveated by the unusual

vegetation stress response.





## 2.2 The Modelling Framework

The CSIRO Chemical Transport Model (CTM) has been developed over 15 years for Australian regional air quality issues (Cope et al., 2004). The CTM is a three-dimensional Eulerian chemical transport model with 35 levels in the vertical to 40km. The CTM has the capability of modelling the emissions, transport, chemical transformation, wet and dry deposition of a

coupled gas and aerosol phase atmospheric system. The modelling uses a nested approach, downscaling from global background concentrations which are advected into the Australian region by the prevailing winds. The Australia-wide domain at 80km resolution is used to simulate the transport of species from large scale continental processes that feed into the boundary conditions of three successively smaller nested grids. The highest resolution grid (3km) has a domain size of 180 x 180 km and is centred on each field campaign site.

The CTM is driven by meteorology from the Conformal Cubic Atmospheric Model (CCAM, (McGregor and Dix, 2008)). CCAM is a global stretched grid dynamical model, used for the prediction of wind velocity, temperature, water vapour mixing ratio (including clouds), radiation and turbulence. CCAM has been evaluated for use in Australia and elsewhere (Corney et al., 2013; Nguyen et al., 2014). CCAM uses the Australian land surface scheme, CABLE (Kowalczyk et al., 2013) to provide information on the surface roughness and leaf area index (LAI, based on MODIS data).

We have included MEGAN as an option in the CTM to calculate the biogenic emissions, the set-up of which is described below. Anthropogenic emissions are based on the Sydney Greater Metropolitan Region inventory (NSW Department of Environment, Climate Change and Water, now NSW EPA (DECCW, 2007)) and includes 37 species. The chemical transformation of gas-phase species is modelled using an extended version of the Carbon Bond 5 mechanism (Sarwar et al., 2008) with updated toluene chemistry (Sarwar et al., 2011). A two-bin sectional scheme calculates the aerosol concentrations,

using the Volatility Basis Set (Shrivastava et al., 2008) for the secondary organic species partitioning, and ISORROPIA (Fountoukis and Nenes, 2007) for the inorganic partitioning. The CTM runs on a chemical timestep of 5 minutes with hourly output of all variables. Table 1 details how the model has been set up and run, along with particulars of the sensitivity runs completed.

## 2.3 Coupling MEGAN to the CSIRO CTM

MEGAN was developed to provide a parameterisation for BVOC emissions and detailed descriptions can be found in Guenther et al. (2012), with a useful review of modules given in Sindelarova et al. (2014). The most recent version, MEGANv2.1 emits 147 species into 19 BVOC classes, which can be output into lumped species appropriate for a number of popular chemical mechanisms, including the Carbon Bond 5 mechanism.

MEGANv2.1 is available as an offline code at http://lar.wsu.edu/megan/guides.html. The code is set-up for use with the

Weather Research and Forecasting (WRF) modelling system, but does not include the effect of $CO_2$ on isoprene as per Heald et al. (2009), nor the effects of soil moisture. In this work, the MEGANv2.1 code has been extracted from the WRF system and coupled to the CSIRO CTM.

MEGANv2.1 provides two approaches for estimating emission factors. The first is to use the 16 plant functional type (PFT) distributions and the global average PFT specific emission factors listed in Table 2 of Guenther et al. (2012). In this case the

emission rate, R ($\mu$g/m$^2$/hr) of species i in any grid box, will be sensitive to the PFT distributions used for the MEGAN simulation (Eq 1):

$$R_i = \sum_{j=1}^{nPFT}\left(EF_{ij} \times \gamma_{ij} \times \chi_j\right) \qquad (1)$$





where $EF_{ij}$ is the emission factor (μg/m$^2$/hr) of species i under standard conditions for PFT j with fractional grid box areal coverage $\chi_j$. The emission activity factor $\gamma_{ij}$ (dimensionless) accounts for emission control processes and uses the following variables to drive the canopy model: compound class, response to light and temperature, leaf age, soil moisture, $CO_2$ and LAI.

The second approach is to use MEGAN global emission factor maps, which are based on plant type composition and plant type specific emission factors. In this case, the MEGAN simulation uses PFTs to define the canopy environment characteristics and to define the fractional grid box areal coverage, but the results are not as sensitive to the PFT data used. The emission rate, R for species i in a given grid cell, xy is (Eq 2):

$$R_i = EF_{i,xy} \sum_{j=1}^{nPFT} (\gamma_{ij} \times \chi_j) \qquad (2)$$

This study uses both approaches, the latter approach for 10 species where emission factor maps are available, and the former approach for all other species. Global emission factor maps for isoprene, myrcene, sabinene, limonene, 3-carene, ocimene, α-pinene, β-pinene, 232-MBO (2-methyl-3-buten-2-ol) and NO are provided at a 1km resolution with the MEGANv2.1 code download, and described below.

### 2.3.1    Production of Emission Factor maps for Australia

The MEGANv2.1 emission factor maps provide values for a specific location based on estimates of plant type composition, which can be individual plant species or more general types, and emission factors for each plant type. The global MEGAN PFT database was used to quantify the fraction of trees, shrubs, crops and herbs at each location in Australia. The tree/shrub type composition for Australia was then determined from data compiled by the Australian Department of Agriculture and Water Resources (DAWR) and released on the data.gov.au data portal in 2003 (URL: http://data.gov.au/dataset/forests-of-australia-2003). The DAWR landcover data are representative of the time period of 1996 to 2002 and include 20 categories. Australia has unusually low tree/shrub genera diversity and many of these landscapes were represented in the DAWR database by a single tree/shrub genera (e.g., *Acacia, Callitris, Casuarina, Eucalyptus, Melaleuca*) although some were more diverse (Mangrove, Rainforest). The landscapes dominated by one genera were assigned the genera average emission factor in the MEGAN plant type database. Mixed landscapes were assigned a representative plant type (e.g., the emission factor for the genera *Avicennia* was assigned to trees in the Mangrove landscape).

The MEGANv2.1 emission factor database classifies Eucalyptus as a high emitter (>10 μg/g/hr), Casuarina and Melaleuca as moderate emitters (1-10 μg/g/hr), and Avicennia and Callitris as very low emitters (<1 μg/g/hr). The Acacia genus includes some high isoprene emitting species that have been identified in Africa (Harley et al., 2003) but the Australian Acacias were assigned a very low isoprene emission rate. The MEGANv2.1 isoprene emission factor for Eucalyptus was based on six enclosure measurement studies (Evans et al., 1982; Winer et al., 1983; Guenther et al., 1991; Street et al., 1997; Loreto and Delfine, 2000; He et al., 2000). Of these studies, only He et al. 2000 was conducted in Australia. These studies report a large range of emission rates that are equivalent to MEGAN landscape emission factors of 1.6 to 51 mg/g/hr. Large variability (more than a factor of 3) was observed for different plants of the same Eucalypt species measured in a single study (Guenther et al. 1991). The average isoprene emission factor of 15 Eucalypt species measured by He et al. 2000, about 24 mg/m$^2$/hr, was similar to the mean value for the other five studies and used as the basis for assigning Eucalypts an isoprene emission factor of 24 mg/m$^2$/hr. The various PFTs listed in Table 2 of Guenther et al. (2012) are comprised of various plant species which includes high, moderate, low and very low emitters. The highest tabulated PFT average emission factor, 11 mg/m$^2$/hr, is assigned to broadleaf deciduous boreal trees, which were determined to be ~46% high emitters (24 mg/m$^2$/hr), based on the MEGAN global landcover dataset.



The distribution of isoprene emission factors in southeast Australia are shown in Figure 1(a). The region between Melbourne and Sydney is covered in vegetation emitting at the upper end of the map scale, close to 24 mg/m$^2$/hr.

### 2.3.2 Meteorological and related inputs to MEGAN

The MEGAN canopy model requires photosynthetically active radiation (PAR), temperature, pressure, relative humidity and
LAI. CCAM supplies hourly temperature and PAR, which exhibit diurnal cycles with early afternoon maxima. The hourly PAR is reduced by a cloud attenuation factor when conditions are cloudy. MEGAN also requires an estimate of previous growing conditions, and needs 24 hour and 240 hour averaged temperature and PAR. The 24 hour average of temperature is provided by CCAM. The 240 hour averaged temperature is fixed at the observed average temperature for the duration of each campaign. The 24 hour averaged PAR is set using measured solar radiation (in W/m$^2$) rather than CCAM output when
measurements were available during the SPS2 and MUMBA campaigns. The observed and modelled PAR from the respective receptor sites are presented in Figure 2. This calculation assumes PAR is half the total solar radiation fraction in the 400 – 700nm wavelength band, and the conversion factor from W/m$^2$ to μmol/m$^2$/s$^1$ is 4.5. The model predicts the correct shape of the diurnal profile but over-predicts by 126 μmol/m$^2$/s$^1$ (7%) at noon during summer (MUMBA) and under-predicts by 236 μmol/m$^2$/s$^1$ (25%) during autumn (SPS2). Average campaign modelled PAR is used for SPS1 and Tumbarumba. Values for
temperature and PAR are given in Table 1.

LAI data is provided from CCAM as described, at the same resolution as each model grid. The distribution of LAI in summer (January) are shown in Figure 1(b), with high LAI data in the region of 5-6 m$^2$/m$^2$ in the coastal plains and mountain ranges of southeast Australia.

### 2.3.3 Construction of high resolution PFT map for Australia

The Community Land Model PFT data from the NCAR data repository is provided on a 0.5° x 0.5° resolution, which when downscaled to the inner 3km grids for the CSIRO-CTM is not suitable (shown in Section 3.2). A new PFT dataset has been constructed for this work, as 3km resolution data in the same format as the 16 PFTs required by MEGAN is not available. A dataset from the International Geosphere Biosphere Project (IGBP) available at a resolution of 1km with 17 landcover types (Belward et al., 1999) was used. The IGBP dataset was converted into NCAR PFTs based on the schemes of Bonan et al.
(2002), Poulter et al. (2011) and local knowledge. Bonan et al. (2002) suggest how much bare ground should be introduced to each PFT grid cell, and also how best to split the boreal from the temperate and tropical plant types using the average temperature of the coldest month. A 30 year climatology of observed average winter temperatures (June - August) in Australia from the Bureau of Meteorology was used for this purpose (www.bom.gov.au/jsp/awap).

Poulter et al (2011) noticed that IGBP classified much of Australia's interior with open shrublands. As a result, 'shrublands',
'grasslands' and 'savannahs' were split into a combination of shrubs and grass as per their implementation in CABLE. Neither Bonan et al (2002) nor CABLE have vegetation occurring within 'urban' landcover types, which would lead to zero biogenic VOC emissions in Sydney within this high resolution implementation. An estimate of vegetation cover in Australian urban areas was made based on Kirstine and Galbally (2004). Table S1 in the supplementary material gives details of how the IGBP landcover dataset was split into the NCAR 16 PFTs suitable for MEGAN. Figure 3 shows the resulting spatial extent of the
PFTs that contribute at the field campaign sites. The maps show a high density (in most cases 95% coverage) of broadleaf evergreen temperate trees around the coastal area. Shrubs and grasslands dominate the north west region, with crops dominating the area in between.





## 3 Results

### 3.1 Contribution of plant functional types to emissions

We calculate the isoprene and monoterpene emission rates per plant functional type for each field campaign's inner nested grids in the model (180 km x 180 km). The SPS and MUMBA grids are coastal and therefore contain a high percentage of zero emitting ocean squares. The bar chart in Figure 4 shows the emission rate for isoprene is an order of magnitude more than monoterpenes, and that broadleaf evergreen temperate trees dominate all campaign airsheds. Tumbarumba is located near an agricultural region and is influenced by emissions from crops, though whether these are croplands or pasture for animals is uncertain. The combination of high emission factors and percentage of broadleaf evergreen temperate trees in the Tumbarumba grid enables up to 3.2 µg/m²/hr of isoprene to be emitted (includes crop PFTs). A sensitivity study conducted for Tumbarumba transferred 50% of the crop area to grassland. This resulted in reducing the peak isoprene by 0.5 – 0.7 ppb, but did not affect the monoterpene concentrations.

### 3.2 Comparisons of modelled and observed BVOCs

Observed and modelled isoprene and monoterpenes are presented as timeseries for the four field campaigns in Figure 5. Modelled isoprene is mostly over-predicted and monoterpenes mostly under-predicted. The model captures the general peaks and troughs in the data, but at the wrong magnitude.

There is missing data from the observed SPS1 dataset and it is not obvious whether observed concentrations would have risen further on 18-19th February 2011 as the model suggests. Also shown on the SPS1 time series (Figure 5 top plots) are the results using the coarse 0.5x0.5 degree resolution PFT map. The very low concentrations of isoprene (peak of 0.2 ppb) show that resolution of the input data is important, and recreating the PFT maps was necessary.

Two of the first three modelled isoprene peaks in the MUMBA dataset (Figure 5 third plots down) coincide with very hot (>40°C) measured days. The first modelled isoprene peak on January 8th is 38 ppb at 43°C, yet the observed peak is 5 ppb at 41°C. There may be isoprene inhibition at temperatures in excess of 40°C which is not represented by the model (Guenther et al., 1991). January 8th is the only day CCAM predicts above 40°C during MUMBA, whilst observations on the 8th and 18th are also above 40°C. CCAM predicts 33°C on the 18th leading to modelled isoprene of 7 ppb; the observations show 4.5 ppb at 44°C. The modelled peak of 8 ppb at 32 °C on January 12th is not mirrored by an observed peak. Whilst temperatures were hot throughout NSW on January 12th, a sea-breeze kept Wollongong cooler at 25°C. The modelled monoterpene Tumbarumba dataset has a number of peaks not seen in the observations (Figure 5 bottom plots).

Figure 6 shows the eddy covariance flux measurements of isoprene and monoterpenes from the post-storm period at Tumbarumba. Uncertainty in the night-time observations are 40% because advection terms were not well constrained, however the daytime fluxes that dominate, are within typical levels of uncertainty. The observed diurnal cycles are compared to modelled emission flux data for the same time period in Figure 6. These observations show peak monoterpene fluxes under 0.8 mg/m²/hr at a time when the monoterpene response increased by a factor of four as a result of the storm (Maleknia et al 2009). Observed isoprene fluxes peak under 0.2 mg/m²/hr. The midday modelled emission rates over-predict the observed isoprene fluxes by a factor of 3, and under-predict the monoterpene fluxes by a factor of 4. Comparing the emission fluxes directly gives confidence that the modelled discrepancy is principally due to the emissions rather than model transport or chemical processes (shown in the supplementary material).





Kanawade et al. (2011) calculated ratios of emitted isoprene to monoterpene carbon for forests in Michigan (ratio = 26.4) and the Amazon (ratio = 15.2)(Greenberg et al., 2004), which are isoprene dominated, whilst forests in Finland (ratio = 0.18) are dominated by monoterpenes (Spirig et al., 2004). These Tumbarumba data show a ratio of 0.14 highlighting the monoterpene dominance after the storm. If the storm had not taken place, we suggest that isoprene and monoterpene emission fluxes would

be broadly similar for both chemical species, but more measurements are needed to confirm this. The magnitudes of the average observed isoprene and monoterpene atmospheric concentrations are broadly similar for all four field studies, shown in Table 2. As atmospheric concentrations are directly related to their emissions rates, the magnitudes of isoprene and monoterpene emission fluxes must be similar under normal (non-storm) conditions, and the ratio of emitted isoprene carbon to monoterpene carbon could be ~0.5-2. This phenomenon may be unique to southeast Australia.

Figure 7 shows campaign average diurnal time series for isoprene, monoterpenes and the ratio of carbon in isoprene versus monoterpene atmospheric concentrations, comparing the CTM to the observations. In most cases the MEGAN scheme predicts the shape of the diurnal profiles well, but isoprene is over-predicted during all four field campaigns. A similar over-prediction in isoprene concentrations occurred using the CHIMERE model, run with MEGANv2.04 at 9km resolution during the MUMBA campaign (Paton-Walsh et al., submitted).

The peak in modelled isoprene is over-predicted by factors between 2 and 6, which will have a flow-on effect through the chemistry via oxidant availability. The modelled isoprene profile captures the observed peak at 10am seen at MUMBA in summer. The observed late afternoon peak in isoprene during SPS2 is diagnosed as due to a collapsing autumnal boundary layer where oxidants at this time are depleted, but isoprene continues to be emitted.

The observed ratio of isoprene carbon versus monoterpene carbon peaks under ~2.5 at all four field studies. The model over-
predicts the observed ratio by factors between 3 and 10, the latter at MUMBA where lower monoterpene concentrations were predicted compared with Sydney and Tumbarumba.

The modelled profile of monoterpenes generally matches the observed peaks for SPS1, SPS2 and MUMBA campaigns, but the magnitude is under-predicted particularly at night by factors between 3 and 4. At Tumbarumba the model predicts a similar monoterpene profile (peaks at night) to the other field campaigns, but the observations show a light dependent profile, similar
to isoprene. This could indicate plant stress due to storm damage occurring that week (Harley et al., 2014). This process is not in the model.

Clearly, modelled isoprene is too high and monoterpenes are too low in southeast Australia. Sensitivity runs are conducted to establish the magnitudes of emission corrections needed to achieve better model/observation agreement. Emission factors for isoprene were reduced by a factor of 3. The emission factors for the monoterpenes species myrcene, sabinene, limonene, 3-
carene, ocimene, α-pinene and β-pinene were increased by a factor of 3.5. Other monoterpene species remain unchanged as their concentrations do not dominate the total (Sindelarova et al., 2014).

The modelled diurnal cycles from the emission factor sensitivity tests are shown as dashed red lines within Figure 7. The reduction in isoprene and increase in monoterpenes show better modelled agreement for most campaigns, but particularly for isoprene in SPS1 and monoterpenes at MUMBA. The ratio of isoprene carbon to monoterpene carbon concentrations from the
emission factor sensitivity test give more reasonable results at MUMBA and Tumbarumba, but under-predict the observed ratio for SPS1 and SPS2. Reducing the isoprene emission factors has incurred a linear response in reducing the isoprene concentrations, but the factor of 3 used is not suitable for all the field campaign data. At Tumbarumba, the reduction is likely a factor of 6. Similarly the monoterpene increase by a factor of 3.5 does not suit all Australian conditions. Nevertheless, these results indicate the magnitude of the corrections required.





Figure 8 shows quantile – quantile plots showing modelled and observed data ranked in ascending order. They highlight any systematic biases that exist in the modelled data; if the modelled data were exactly like the observations then the points would sit on the 1:1 line. Figure 8 shows the 1:1 line with two dashed lines representing a factor of 2 either side. The aim is to further examine the extent of the over/under-prediction in isoprene and monoterpenes. The data is paired; if the PTR-MS was offline then the modelled data is removed for these times. The normalised mean bias is calculated; values closer to zero exhibit less bias.

There is a large model over-prediction in isoprene and therefore the isoprene products. Note that measurements of isoprene products were not made available from Tumbarumba. The modelled monoterpenes are under-predicted by just over a factor of 2 in most cases. The one exception is Tumbarumba which has zero model bias in monoterpenes, however the shape of the modelled diurnal cycle was at odds with the observed profile. The results from the emission factor sensitivity test show better modelled isoprene profiles, but the factor of 3.5 increase in monoterpene emissions is too high. The bias in modelled VOCs is reduced in the emission factor sensitivity test. For isoprene the bias switches from positive to negative indicating the chosen decrease factor is too high. The increase factor for monoterpenes is too high for SPS1 and SPS2, both of which show equal sized biases but with the opposite sign to the bias in the base case run.

The concentrations of the isoprene products can also be used to evaluate the lifetime of isoprene in the model and observations. Figure 9 shows the ratio of isoprene and its products to the isoprene products. This examines whether the model chemistry is proceeding at observed rates. The results show high correlations >0.85 for the observed ratios; correlations in excess of 0.90 for SPS1 and SPS2 for species modelled by the base case run; and less well correlated (>0.78) in the modelled base case at MUMBA. More isoprene products are predicted by the model than the observations for SPS1. This suggests that oxidation occurs faster in the model compared to the observations for February 2011. However the modelled rates of oxidation are more reasonable for SPS2 and MUMBA. There is a slight improvement in the $r^2$ correlation coefficient between species modelled by the emission factor sensitivity test for SPS1 and SPS2.

## 4 Summary and Conclusions

MEGANv2.1 has been incorporated into the CSIRO Chemical Transport Model. The CTM used a nested grid approach, downscaling from an Australia wide grid to focus on receptor sites at a resolution of 3 km. This high resolution approach required a new plant functional type map to be constructed for Australia from an IGBP 1km dataset. Whilst deconstructing the IGBP dataset to fit the NCAR PFTs has been done in accordance with literature and local knowledge, it is subjective and may have introduced uncertainty. The model was used to predict concentrations measured during four field campaigns in southeast Australia; one in spring (Tumbarumba), two in summer (SPS1 and MUMBA) and one in Autumn (SPS2). The observed concentrations of isoprene and monoterpenes were of a broadly similar magnitude, which may indicate that southeast Australia holds an unusual position where neither chemical species dominates. The model over-predicted isoprene concentrations (up to a factor of 6) and under-predicted monoterpenes (up to a factor of 4). A short series of measured emission fluxes at Tumbarumba showed that the model over-predicted isoprene fluxes by a factor of 3 and under-predicted monoterpene fluxes by a factor of 4 at midday.

Southeast Australia is dominated by forested regions, and cities here are surrounded by a high source of BVOC emissions. These BVOCs have the capacity to dominate atmospheric chemical processes in urban airsheds during the high temperatures experienced in Australian summers. Southeast Australia has been considered a global hotspot for isoprene emissions due to the presence of high emitting Eucalypt species (Guenther et al., 2012) although our results indicate that Eucalypts may not



emit as much isoprene as previously thought. The MEGANv2.1 isoprene and monoterpene emission factors assigned to Eucalypts, 24 and 1.6 mg/m²/hr respectively, are higher than the global average value of all broadleaf evergreen temperate trees (10 and 0.99 mg/m²/hr, table 2 of Guenther et al., 2012) because not all broadleaf evergreen temperate trees have high isoprene and monoterpene emissions.

While there is a limited understanding of all of the processes controlling biogenic VOC emissions, such as the impact of droughts, which can lead to an inhibition of BVOC emissions (Sharkey and Loreto, 1993; Pegoraro et al., 2007), the overall emission can be adjusted by revising the emission factor. A sensitivity study reduced the emission factors of isoprene by 3 and increased the monoterpene emission factors by 3.5. The effects on the modelled concentrations was roughly linear. This experiment showed that there is no single increase/decrease factor which suits all locations/seasons found in southeast
Australia, indicating that adjustment is needed not only in the emission factors but also in the representations of the processes controlling emissions variations.

The MEGANv2.1 emission factors for Eucalyptus were primarily based on enclosure measurements of young trees. Street et al. (1997) conducted field enclosure measurements of *Eucalyptus globulus* trees in a plantation in Portugal and found that both isoprene and monoterpene emissions from a 7 year old tree were about 5 times lower than the emissions of a year old sapling.
Nunes and Pio (2001) compared emissions from two year old *Eucalyptus globulus* saplings in the laboratory and 7 year old trees in a plantation and found the adult tree isoprene emissions were about a third lower than that of the young tree. The isoprene emission rates of adult *E. globulus, E. grandis* and *E. camaldulensis* trees measured by Winter et al. (2009) are a factor of four lower than the emissions that He et al. (2000) measured from 2 year old potted saplings of the same three Eucalypt species. This is in good agreement with the results of Street et al. (1997) and Nunes and Pio (2001). The monoterpene
emission rates measured by Winter et al. (2009) for adult trees, however, were a factor of four higher than the 2 year old saplings measured by He et al. (2000). This does not agree with the findings of Street et al. (1997), but does agree with the higher than predicted atmospheric concentration measured at the field sites described in this paper. These results suggest that the MEGANv2.1 isoprene emission factors are biased by being based on measurements of young trees and should be decreased by up to a factor of four or five considering that the isoprene emitting canopy consists primarily of adult trees. This would
result in better agreement with the observed ambient isoprene concentrations described above. The results of monoterpene enclosure studies are more inconclusive and are also difficult to interpret due to artefacts associated with elevated emissions from disturbance of the monoterpene storage structures (Winters et al. 2009).

In order to more accurately characterize the atmospheric chemistry, air quality and climate in Australia, further observations and quantitative analysis of Australian BVOC emission rates are needed. Australia is biologically diverse and the canopy and
understory are composed of many other species in addition to Eucalypts. Satellite column measurements of BVOC oxidation products such as formaldehyde and glyoxal are available and can be useful for investigating regional and seasonal distributions of biogenic emissions (Palmer et al., 2003; Kaiser et al., 2015). Direct flux measurements, using towers and aircraft eddy flux approaches, are needed to provide a direct constraint on Australian BVOC emissions (Karl et al., 2013).

**Acknowledgements**

The NSW Environment Trust has provided support for this study through the "Atmospheric Particles in Sydney: model-observation verification study". CPW wishes to acknowledge the Australian Research Council for funding as part of the Discovery Project DP110101948. KME wishes to thank R. Law for helpful discussions on plant functional types in Australia;



to P. Hoffman for assistance with CDO; all those from CSIRO and the University of Wollongong for fieldwork participation; and to J. Fisher and L. Emmons for helpful comments on the manuscript.

The LAI data product was retrieved from MCD15A2 version 4 from the online Data Pool, courtesy of the NASA Land Processes Distributed Active Archive Center (LP DAAC), USGS/Earth Resources Observation and Science (EROS) Center,

5   Sioux Falls, South Dakota, https://lpdaac.usgs.gov/data_access/data_pool.





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

30

35



Table 1 Model set-up and list of model runs completed

|  | SPS1 | SPS2 | MUMBA | Tumbarumba |
|---|---|---|---|---|
| 240 hour average Temperature, K | 295 | 290 | 295 | 289 |
| 24 hour average PAR, $\mu$mol m$^{-2}$ s$^{-1}$ | 437 | 305 | 485 | 500 |
| Coarse grid PFT | X |  |  |  |
| Base MEGAN run | X | X | X | X |
| Exchange 50% crops → grass |  |  |  | X |
| Emission factors isoprene /3 monoterpenes x3.5 | X | X | X | X |
| ± 20% NOx emissions* | X | X | X | X |

* Shown in supplementary material.





Table 2 Average (min-max) observed isoprene and monoterpene concentrations at all four field sites.

| Observations | Isoprene | Monoterpenes |
|---|---|---|
| | ppb | ppb |
| SPS1 | 0.76 | 0.44 |
| | (0.09* -7.10) | (0.20* -2.74) |
| SPS2 | 0.63 | 0.46 |
| | (0.01-4.63) | (0.006-1.95) |
| MUMBA | 0.28 | 0.12 |
| | (0.002-4.57) | (0.004*-1.39) |
| Tumbarumba | 0.15 | 0.20 |
| | (0.02*-1.01) | (0.02*-1.79) |

* values equate to half the limit of detection.





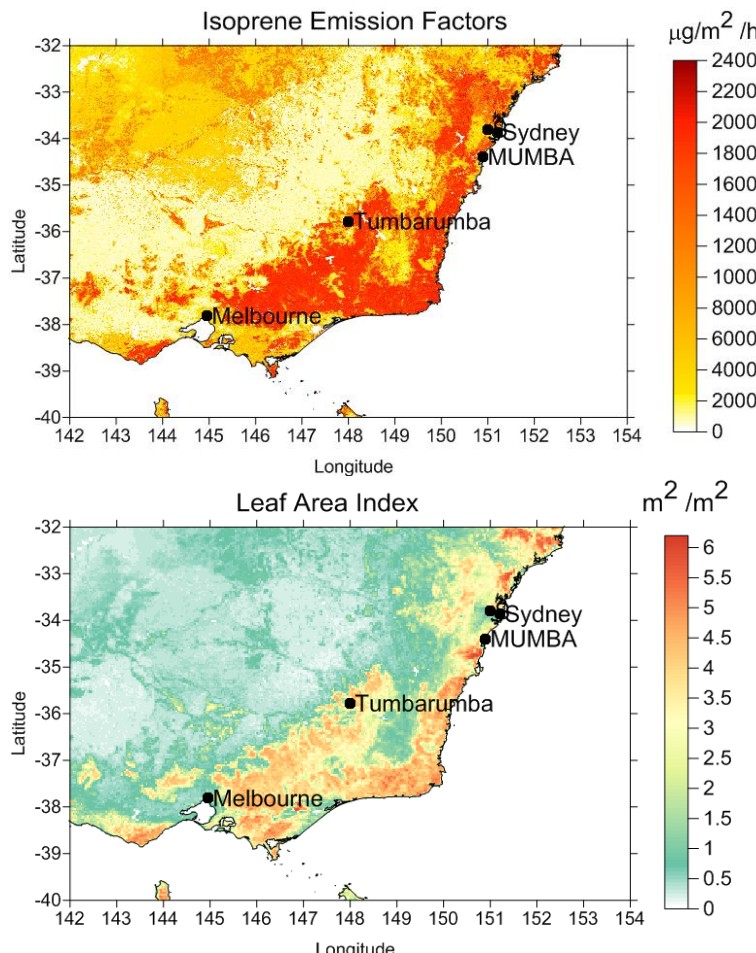

Figure 1 Southeast Australia at 1km resolution, showing (a) Isoprene from the MEGAN global emission factor map and (b) LAI in January. Field campaign locations are also shown with locations of major cities. The Sydney field campaigns were located west of Sydney marker.





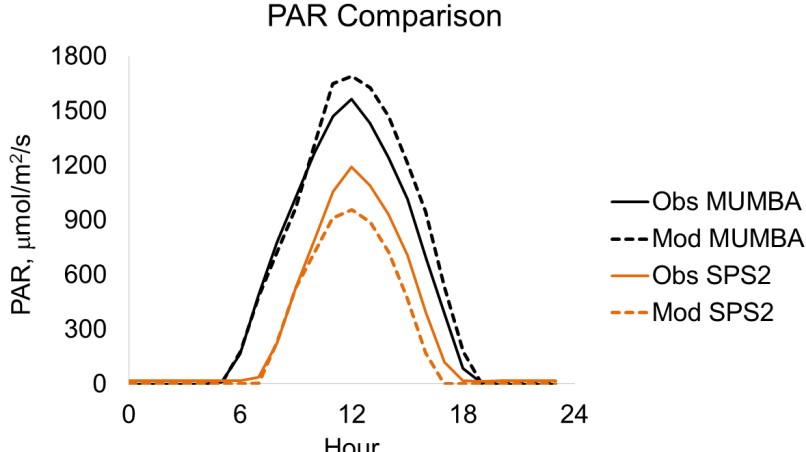

Figure 2 Comparison of photosynthetically active radiation for modelled and measured SPS2 and MUMBA data.



Figure 3 The percentage area covered by the indicated PFTs resulting from splitting the 1km IGBP database into NCAR PFTs in southeast Australia.



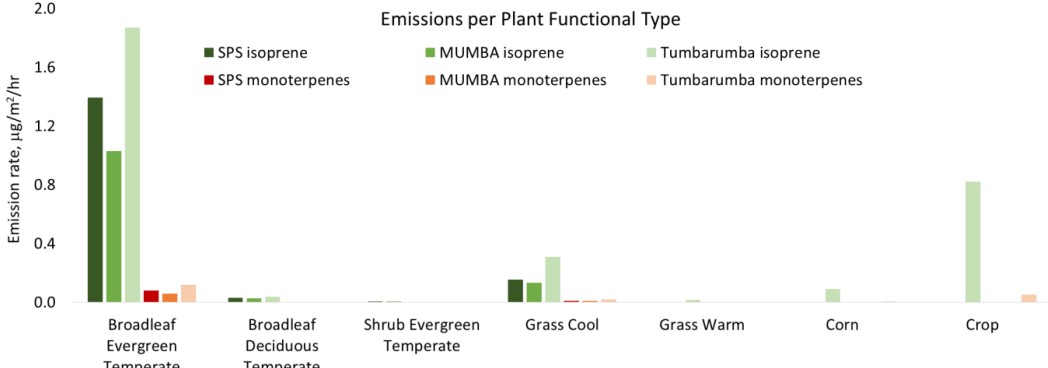

Figure 4 Emission rates of isoprene and monoterpenes per PFT within each campaign's inner nested grid (180km x 180 km).





Figure 5 Time series of observed and modelled isoprene (left) and monoterpenes (right) for each field campaign. SPS1 = blue, SPS2 = red, MUMBA = yellow, Tumbarumba = green. Y-axis for isoprene during MUMBA restricted to 10ppb, as modelled peak is 38 ppb on 8.1.13.





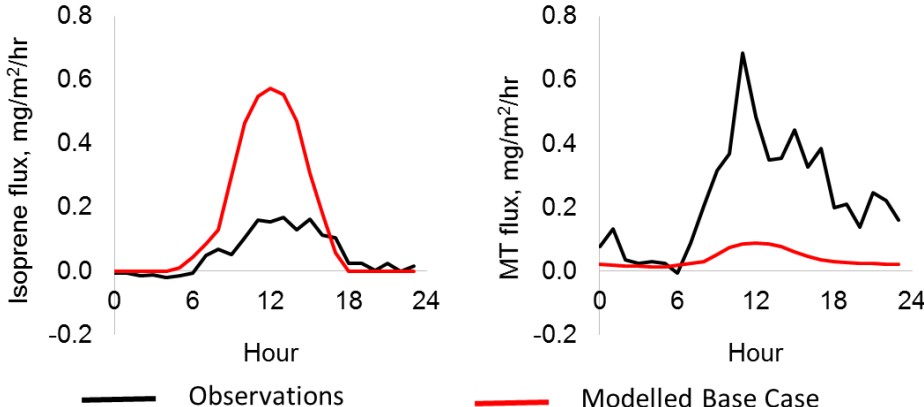

Figure 6 Diurnal cycles of isoprene (left) and monoterpene (MT, right) emission fluxes from three days of eddy covariance measurements at Tumbarumba during November 2006. Modelled emission fluxes are plotted from the same time period.





Figure 7 Campaign average diurnal cycles for isoprene (left), monoterpenes (middle) and the ratio of isoprene

carbon to monoterpene (MT) carbon (right). S1=SPS1, S2=SPS2, M=MUMBA T=Tumbarumba. F2= percentage

within a factor of 2 between observations and base run.



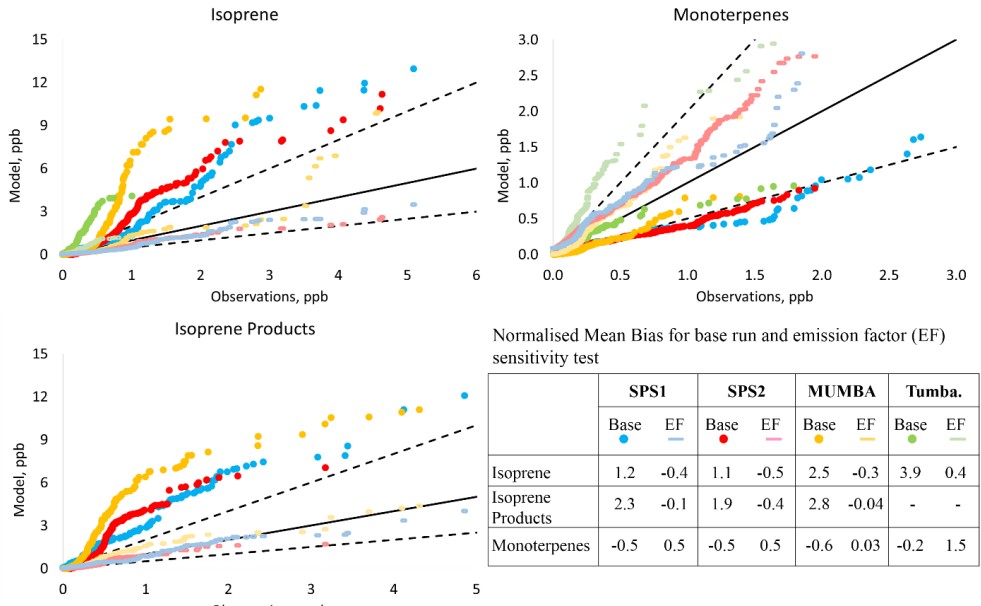

2   Figure 8 Quantile-quantile plots to show relationship between modelled and observed biogenic gases. The base

3   run are dots, the emission factor sensitivity study are the dashes. The solid black line = 1:1; dashed black lines

4   indicate ± a factor of 2. Note: isoprene products are MVK + MACR. The y-axis on isoprene chart is reduced to 15

5   ppb to aid visual comparison, as modelled MUMBA data reaches 38 ppb.





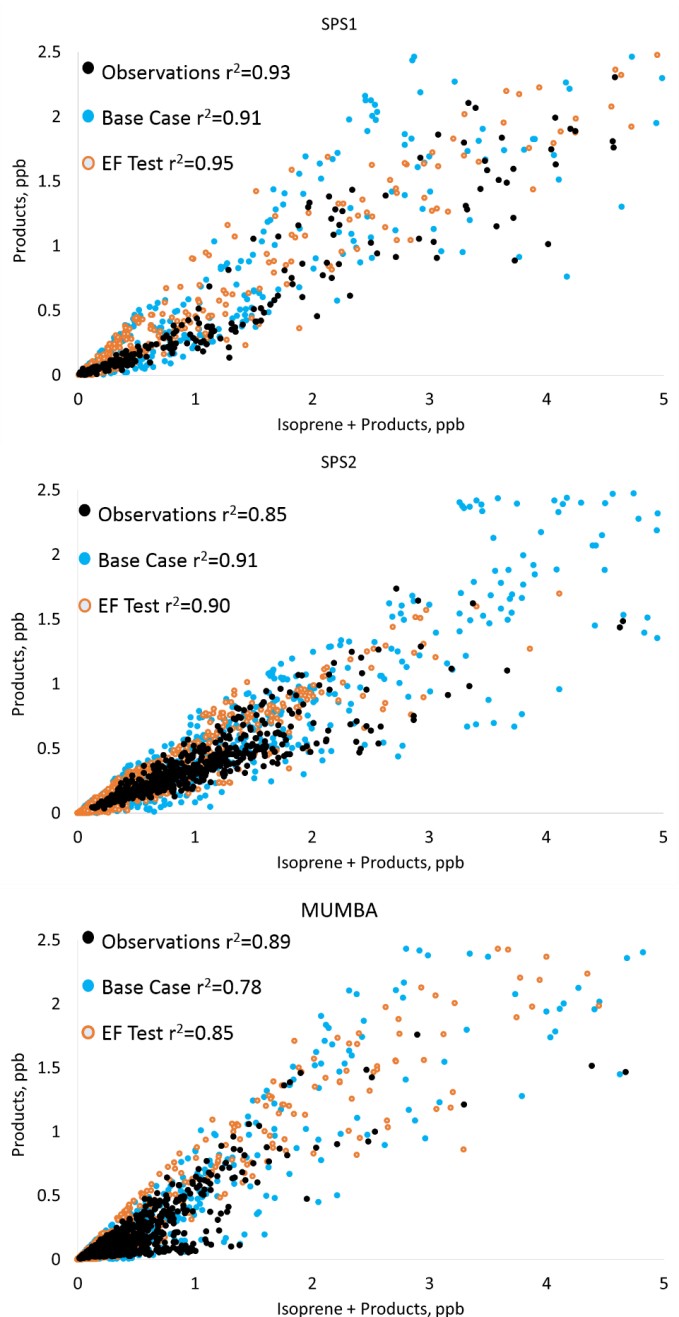

2      Figure 9 Scatterplots of modelled and observed ratios between isoprene and the isoprene products, with $r^2$

3      correlation coefficients. EF = emission factor sensitivity test. Note, x and y axes restricted to 5 ppb and 2.5 ppb

4      respectively.