# Peer review of "Current estimates of biogenic emissions from Eucalypts 1"

_Atmospheric Chemistry and Physics, 2016_

## Referee Comment (RC1) · Anonymous Referee #1 · 26 Apr 2016

Emmerson et al. presents the first detailed evaluation of the biogenic emissions model MEGAN coupled to a high resolution regional tropospheric chemistry model over southeast Australia. Biogenic VOC emissions are a significant portion of emissions in the Southern Hemisphere and the MEGAN model indicates southeast Australia has one of the highest isoprene-emitting regions. The regional model (CCAM) was run at 3km horizontal resolution, requiring the development of a new 3km landcover and vegetation map with which to drive MEGAN, which was done by combining results from several previous studies. Observations of ambient mixing ratios and emissions fluxes of isoprene and terpenes were used from several studies in southeast Australia were used for a quantitative evaluation of the emissions. The indication that isoprene and terpenes are emitted in comparable amounts is a new finding. The isoprene oxidation scheme in the regional model CCAM is also evaluated by comparison to observed isoprene oxidation products (MVK and methacrolein). In addition to the specific results of this study, this paper illustrates a procedure for quantitative evaluation of biogenic emissions models that should be applied to other parts of the globe.

This paper is clearly written and the figures clearly illustrate the conclusions of the paper. The title and abstract accurately represent the results of the paper. I recommend publication of this paper after addressing the minor comments listed below.

Intro. p1,l.29: Say 'approximately' instead of 'in the region of' because you also talk about a lot of geographical regions.

p.3, l.32: 'emission pattern' is confusing. Isn't it just 'emissions'?

p.8, l.16: 'via' - do you mean 'dependent on'? Perhaps re-word 'flow-on' also.

p.9,l.26: 'domain' instead of 'grid'; 'simulation' instead of 'approach'

p.10, l.23: add 'for Eucalypt' after 'emission factors'

---

## Referee Comment (RC2) · Anonymous Referee #2 · 30 Apr 2016

Review of Emmerson et al., "Current estimates of biogenic emissions from Eucalypts uncertain for Southeast Australia"

Synopsis

Emmerson et al., report on a study to determine the emission factors of isoprene and monoterpenes by vegetation in Southeast Australia, as measured during a number of field campaigns and modelled using the MEGAN emission module coupled to the CSIRO Chemical Transport Model. They identify large discrepancies between measured and modelled emission fluxes, which they attribute to overestimated isoprene emission factors used in MEGAN. They give suggestions on how to improve this, but caution that they did not find a single change that would improve predictions under all conditions investigate, and call for further observations to better the situation.

General remarks

The paper is clearly written and well structured. Methods and results are presented in a concise manner, and the conclusions drawn are sound. I did not find anything of concern while conduction my review and hence recommend for publication after the two following minor comments are addressed:

Detailed comments

P 2 L 26: The effect of soil moisture on plant emissions seems to be an unknown which could potentially have considerable influence on predictions. Even if the model would include it - how good is the soil moisture in the model?

p 4 l 18: CB05 is almost 10 years old now, and our knowledge on isoprene chemistry has improved considerably - IEPOX formation, ISOPOOH and the associated OH recycling directly impact the influence of isoprene on O3, and hence your evaluation. Can you assess how well CB05 performs compared to other mechanisms with a more updated isoprene chemistry? At least mention this potential source of error.

p 8 l 19: Do you have any evaluation of the boundary layer height performance of the modeling system? The modelled concentrations are highly sensitive to this parameter, and especially its dynamic behavior (i.e. the collapse at dusk) can easily be wrong in the model.

Figures: Is it possible that the figures are copy-pasted from Excel or similar? Please improve their quality (spurious frames around them, resolution) to publication standards.

---

## Referee Comment (RC3) · Anonymous Referee #3 · 11 May 2016

This paper investigates the biogenic isoprene and monoterpene emissions in Southeastern Australia using the emissions model, MEGAN, and regional chemical transport models. The model output concentrations, and in one case fluxes, are compared to observations made at four field campaigns in the region. Overall, this is a valuable exercise that highlights the uncertainties in the biogenic emission estimates in Australia and highlights the lack of information needed to constrain the current models. This study is relevant to the readers of ACP and is appropriate for publication in the journal. I recommend that this paper be published with minor edits. I provide my detailed comments here.

In the introduction, the MEGAN version 2.1 paper, Guenther et al. 2012, is often referenced (for example, page 1 line 32 and page 2 line4). However, there are earlier papers that introduce the ideas discussed that should be included (i.e., earlier Guenther et al. papers from the 1990's and the MEGAN version 1 paper, Guenther et al. 2006.).

On lines 23 of page 2, Muller et al. (2008) found overestimates of isoprene. How was this determined, and with what observations?

The outline of the high resolution model grids would be interesting to see on Figure 1.

Page 5: Why were the Acacia species in Australia assigned the lower emission rates?

Page 6, Section 2.3.3: The authors develop a high resolution PFT emission factor map specific to Australia based on an IGBP land cover dataset. Why was this land cover map used? It seems very old, and there are many other more recent land cover datasets available? And is this consistent with the land cover/land use datasets applied in the chemical transport models?

Page 7, line 6: Is the broadleaf evergreen temperate tree PFT in the study dominated by Eucalypts?

Page 8: The authors perform a sensitivity test on the emissions rates. Why (or how) were the factors of 3 for isoprene and 3.5 for monoterpenes chosen? (Lines 27-30).

Figure 1: Which version of MEGAN emission factors are shown here?

Editorial Comments

Page 2, lines 1 and 2: The sentences should read: "all of these processes"

Page 3, line 17: I suggest rewording this sentence: "Two intensive field campaigns took place: SPS1 occurred between . . ."

Page 4, line 30: Remove "as" before "per"

Page 5, line 35: I suggest rewording this sentence: "The PFTs listed in Table 2 of Guenther et al (2012) are comprised of various plant species that include high, moderate, low and very low emitters." I am not sure what the point is of the following sentence,

and this could be removed.

Page 7, line 30: remove the comma after "dominate"

Page 8, lines 1-2: The wording of this should be changed so that the references identified are properly cited. For example: "Calculated ratios of emitted isoprene to monoterpene carbon were found to be 26.4 for forests in Michigan (Kanawade et al. 2011) and 15.2 in the Amazon (Greenberg et al. 2004).

"Data" are plural (i.e., page 7, line 16; page 9, line 4)

---

## Author Comment (AC1) · 23 May 2016

Thank-you for reading the paper and submitting one of the best reviews I've ever received. I would be happy to include all your suggestions to improve the understanding of the article.

Page 8, (now line 19) has been changed to read: "The peak in modelled isoprene is over-predicted by factors between 2 and 6, which will have an effect through the chemistry dependent on oxidant availability. The modelled isoprene profile captures the observed peak at 10am..."

---

## Author Comment (AC2) · 23 May 2016

Thanks to reviewer #2 for your comments. Reviewer #2 touches on a couple of items I have thought a great deal about; choice of chemistry scheme and the boundary layer height in the model.

Detailed comments: P 2 L 26: The effect of soil moisture on plant emissions seems to be an unknown which could potentially have considerable influence on predictions. Even if the model would include it - how good is the soil moisture in the model?

There are two issues here: the soil moisture code within MEGAN and the soil moisture parameter which enters the CTM from the meteorological component (CCAM). This particular version of MEGAN v2.1 returns a value of 1 for Gamma(soil), thus the soil moisture does not influence the BVOC emission. Soil moisture within CCAM comes

from CABLE, the outputs of which could be coupled to MEGAN in future, subject to aligning soil types and textures. Soil moisture at 1cm, 16cm and 44cm are used. CABLE soil moisture within the ACCESS GCM has been assessed by comparison with 19 other models in the CMIP5 evaluation and found to lie at the median of the model ensemble mean (personal communication, Ian Watterson, CSIRO). The CABLE terrestrial water cycle has been evaluated in the global soil wetness project GSWP-2 and found to compare well with evapotranspiration and runoff measurements (Zhang et al., 2013).

The following text has been added at page 4 line 14.

"...CABLE to provide information of surface roughness, soil moisture and leaf area index (LAI, based on MODIS data). The soil moisture parameter has been evaluated indirectly within the Global Soil Wetness Project, by comparing model evapotranspiration and runoff to measurements (Zhang et al., 2013). Whilst CABLE performed well, soil moisture remains a source of uncertainty."

text added at page 4 line 37: "Note that soil moisture is used elsewhere in the CTM to calculate the dust emission flux, and could be coupled with MEGAN in the future".

p 4 l 18: CB05 is almost 10 years old now, and our knowledge on isoprene chemistry has improved considerably - IEPOX formation, ISOPOOH and the associated OH recycling directly impact the influence of isoprene on O3, and hence your evaluation. Can you assess how well CB05 performs compared to other mechanisms with a more updated isoprene chemistry? At least mention this potential source of error.

I have compared CB05's predecessor, CBIV to another five mechanisms, one of which was the Master Chemical Mechanism, and am aware of the differences the choice of mechanism can make to secondary species such as O3 (Emmerson and Evans, 2009). More recently Knote et al (2015) compared a couple of variants of the CB05 mechanism to other schemes containing improved isoprene oxidation schemes. The choice of mechanism resulted in differences in ozone and isoprene concentrations,

particularly in biogenic regions. However, neither the Knote nor Emmerson papers went so far as to compare with measurements nor make an assessment of which scheme was 'best'. I am looking into adapting the most recent version of MOZART into the CSIRO CTM as I would like to calculate MVK and MACR separately, and consider the more recent research into isoprene oxidation pathways particularly the isoprene nitrates.

I will add the following text to page 4 line 21. "The CB05 mechanism treats the production of a lumped isoprene oxidation product only, simplifying the chemistry. More recent schemes consider explicit oxidation products which can affect the production of ozone and nitrate species. The CB05 mechanism and its predecessor CBIV, have been compared with other schemes in Emmerson and Evans (2009) and Knote et al. (2015), but not against measurements. Choice of chemistry scheme can introduce uncertainty, which could be explored in future work".

p 8 l 19: Do you have any evaluation of the boundary layer height performance of the modeling system? The modelled concentrations are highly sensitive to this parameter, and especially its dynamic behavior (i.e. the collapse at dusk) can easily be wrong in the model.

We do not have any measurements of boundary layer height for these field campaigns. We included the ratios of isoprene and the isoprene products at figure 9 with the observations as this exercise removes the dilution effects, and still compared well.

However, we have looked at vertical potential temperature profiles from aircraft taking off from Sydney airport (AMDAR data) as a proxy to compare the model with. The modelled and observed potential temperature profiles compare reasonably well. However, the aircraft take off towards the sea and there is significant horizontal transport of the plane between the readings. The boundary layer can be inferred from these plots by eye, but we found this too subjective. We could also infer the dilution of the atmosphere by inserting a radon source to the model and comparing with radon measurements for

SPS1, SPS2 and MUMBA. This is something I intend to do in future.

The following text has been added to the supplementary section, page 3 line 14.

"There are no direct measurements of boundary layer height for these field campaigns. The model boundary layer height has been compared with vertical potential temperature profiles from aircraft taking off from Sydney airport (AMDAR, http://www.wmo.int/pages/prog/www/GOS/ABO/AMDAR/AMDAR_System.html). From a small sample, the overall profiles compare reasonably well (not shown). However, the aircraft generally take off towards the sea and there is significant horizontal displacement of the plane between the potential temperature readings. We assess that horizontal gradients in temperature and boundary layer height in this coastal region considerably confuse the issue of resolving the boundary layer depth at Westmead, a site 33 km inland. Thus at this stage boundary layer height verification is not possible."

Figures: Is it possible that the figures are copy-pasted from Excel or similar? Please improve their quality (spurious frames around them, resolution) to publication standards.

Agreed, I will work on improving the resolution of the images.

References

Emmerson, K. M., and Evans, M. J.: Comparison of tropospheric gas-phase chemistry schemes for use within global models, Atmos Chem Phys, 9, 1831-1845, DOI 10.5194/acp-9-1831-2009, 2009.

Knote, C., Tuccella, P., Curci, G., Emmons, L., Orlando, J. J., Madronich, S., Baro, R., Jimenez-Guerrero, P., Luecken, D., Hogrefe, C., Forkel, R., Werhahn, J., Hirtl, M., Perez, J. L., San Jose, R., Giordano, L., Brunner, D., Yahya, K., and Zhang, Y.: Influence of the choice of gas-phase mechanism on predictions of key gaseous pollutants during the AQMEII phase-2 intercomparison, Atmos Environ, 115, 553-568, 10.1016/j.atmosenv.2014.11.066, 2015.

Zhang, H. Q., Pak, B., Wang, Y. P., Zhou, X. Y., Zhang, Y. Q., and Zhang, L.: Evaluating Surface Water Cycle Simulated by the Australian Community Land Surface Model (CABLE) across Different Spatial and Temporal Domains, J Hydrometeorol, 14, 1119-1138, 10.1175/Jhm-D-12-0123.1, 2013.

---

## Author Comment (AC3) · 23 May 2016

Thank you to reviewer #3. I am happy to include all your editorial comments and respond to your detailed questions below.

In the introduction, the MEGAN version 2.1 paper, Guenther et al. 2012, is often referenced (for example, page 1 line 32 and page 2 line4). However, there are earlier papers that introduce the ideas discussed that should be included (i.e., earlier Guenther et al. papers from the 1990's and the MEGAN version 1 paper, Guenther et al. 2006.).

I have included the Guenther et al 2006 and Guenther et al 1995 references to page 2 line 5.

On lines 23 of page 2, Muller et al. (2008) found overestimates of isoprene. How was this determined, and with what observations?

Muller used MEGAN v2 and compared the modelled formaldehyde column to GOME satellite observations.

I have rewritten the sentence at page 2 line 23 to read:

"Muller et al. (2008) found an overestimate of isoprene across northern Australia by comparing MEGANv2 to GOME satellite measurements of formaldehyde, and in subsequent work estimated the magnitude of this over-prediction to be a factor of 2-3 in January (Stavrakou et al., 2015)"

The outline of the high resolution model grids would be interesting to see on Figure 1.

Done.

Page 5: Why were the Acacia species in Australia assigned the lower emission rates?

Acacia species in Australia were assigned low isoprene and monoterpene emission rates in MEGAN because the only studies we know of in the scientific literature, which have been exclusively focused on African and North American Acacias, indicate that non-negligible isoprene and monoterpene emission does occur but it is exceptional with only one high monoterpene emitter and one high isoprene emitter reported for the eight species studied. Also, Rei Rasmussen (personal communication) has investigated isoprene emission from some common Australian Acacias and did not find any of them to be isoprene emitters.

I have altered the text on page 5 line 31 to say:

"Isoprene or monoterpene emissions have not been published for any Australian Acacias but eight Acacia species from South Africa (Guenther et al., 1996; Harley et al., 2003) and the US (Guenther et al., 1999; Papiez et al., 2009) have been investigated and only one isoprene emitter and one monoterpene emitter have been identified.

Based on these observations, the MEGAN model assumes low isoprene and monoterpene emission rates for Australian Acacia species."

Page 6, Section 2.3.3: The authors develop a high resolution PFT emission factor map specific to Australia based on an IGBP land cover dataset. Why was this land cover map used? It seems very old, and there are many other more recent land cover datasets available? And is this consistent with the land cover/land use datasets applied in the chemical transport models?

The Bonan et al 2002 paper was a good place to start as they showed a method to directly convert IGBP landcover into the 16 PFT classes required by the MEGAN model. It was the simplest thing to do once it became evident that the coarse resolution PFT global maps would not be suitable. I have added the following to the supplementary section, page 1 line 16:

"When emission factor maps are used, as is the case for the major biogenic species isoprene and a- and b-pinene, the emission rates are not particularly sensitive to this PFT map. Testing the CSIRO-CTM without the emission factor maps would increase the sensitivity to PFT, which could be tested in future work. This could also be a good opportunity to test alternative land cover datasets".

Page 7, line 6: Is the broadleaf evergreen temperate tree PFT in the study dominated by Eucalypts?

Yes, Tumbarumba is surrounded by Eucalypt species of E. delagatensis (Alpine Ash) and E. dalrympleana (Mountain Gum) as described in the field campaign section on page 3 line 29.

I have altered page 7 line 13 to read: "The combination of high emission factors and percentage of broadleaf evergreen temperate trees in the Tumbarumba grid (Eucalypts, section 2.1.3) enables up to 3.2 ug/m2/h of isoprene to be emitted"

Page 8: The authors perform a sensitivity test on the emissions rates. Why (or how)

were the factors of 3 for isoprene and 3.5 for monoterpenes chosen? (Lines 27-30).

The factors are somewhat arbitrary, and chosen by comparing the modelled isoprene and monoterpene diurnal cycles with the observations. The increase/decrease factors varied enormously across the campaigns, however the observed monoterpene profile at Tumbarumba was ignored because it was different to the other measured monoterpene profiles. A decrease factor of 3 for isoprene suited SPS1 best whilst an increase of 3.5 suited the MUMBA monoterpenes profile best.

The text has been updated on page 8 line 33:

"The factors chosen are somewhat arbitrary. A decrease factor of 3 for isoprene suited the SPS1 profile best whilst an increase of 3.5 suited the MUMBA monoterpenes profile best."

Figure 1: Which version of MEGAN emission factors are shown here? The MEGAN emission factor maps are listed as version 2011 and dated 20 March 2013. I have added the following text to page 5 line 15: "(version 2011)"

Editorial Comments

I have made all the changes suggested in this section by reviewer #3

Page 2, lines 1 and 2: The sentences should read: "all of these processes"

Page 3, line 17: I suggest rewording this sentence: "Two intensive field campaigns took place: SPS1 occurred between . . ."

Page 4, line 30: Remove "as" before "per"

Page 5, line 35: I suggest rewording this sentence: "The PFTs listed in Table 2 of Guenther et al (2012) are comprised of various plant species that include high, moderate, low and very low emitters." I am not sure what the point is of the following sentence, and this could be removed.

We are trying to highlight how high the isoprene emission factor assigned to Australian Eucalypts (ie using approach 1 (page5, equation 1) where the model is not sensitive to PFTs) is compared to approach 2 (PFT sensitive). I have deleted the sentence and reworded to:

". . .assigning Eucalypts an isoprene emission factor of 24 mg/m2/hr. This is more than double the isoprene emission factor used for broadleaf evergreen temperate trees if approach 2 is used (PFT sensitive)."

Page 7, line 30: remove the comma after "dominate"

Page 8, lines 1-2: The wording of this should be changed so that the references identified are properly cited. For example: "Calculated ratios of emitted isoprene to monoterpene carbon were found to be 26.4 for forests in Michigan (Kanawade et al. 2011) and 15.2 in the Amazon (Greenberg et al. 2004).

"Data" are plural (i.e., page 7, line 16; page 9, line 4)

References:

Guenther, A., Archer, S., Greenberg, J., Harley, P., Helmig, D., Klinger, L., Vierling, L., Wildermuth, M., Zimmerman, P., and Zitzer, S.: Biogenic hydrocarbon emissions and landcover/climate change in a subtropical savanna, Phys Chem Earth Pt B, 24, 659-667, Doi 10.1016/S1464-1909(99)00062-3, 1999.

Harley, P., Otter, L., Guenther, A., and Greenberg, J.: Micrometeorological and leaf-level measurements of isoprene emissions from a southern African savanna, J Geophys Res-Atmos, 108, Artn 8468 doi:10.1029/2002jd002592, 2003.

Muller, J. F., Stavrakou, T., Wallens, S., De Smedt, I., Van Roozendael, M., Potosnak, M. J., Rinne, J., Munger, B., Goldstein, A., and Guenther, A. B.: Global isoprene emissions estimated using MEGAN, ECMWF analyses and a detailed canopy environment model, Atmos Chem Phys, 8, 1329-1341, 2008.

Papiez, M. R., Potosnak, M. J., Goliff, W. S., Guenther, A. B., Matsunaga, S. N., and Stockwell, W. R.: The impacts of reactive terpene emissions from plants on air quality in Las Vegas, Nevada, Atmos Environ, 43, 4109-4123, 10.1016/j.atmosenv.2009.05.048, 2009.

Stavrakou, T., Muller, J. F., Bauwens, M., De Smedt, I., Van Roozendael, M., De Maziere, M., Vigouroux, C., Hendrick, F., George, M., Clerbaux, C., Coheur, P. F., and Guenther, A.: How consistent are top-down hydrocarbon emissions based on formalde-hyde observations from GOME-2 and OMI?, Atmos Chem Phys, 15, 11861-11884, 10.5194/acp-15-11861-2015, 2015.

---

## Author Response (AR1)

Response to Reviewer #1

This paper is clearly written and the figures clearly illustrate the conclusions of the paper. The title and abstract accurately represent the results of the paper. I recommend publication of this paper after addressing the minor comments listed below.

Intro. p1,l.29: Say 'approximately' instead of 'in the region of' because you also talk about a lot of geographical regions.
p.3, l.32: 'emission pattern' is confusing. Isn't it just 'emissions'?
p.8, l.16: 'via' - do you mean 'dependent on'? Perhaps re-word 'flow-on' also.
p.9,l.26: 'domain' instead of 'grid'; 'simulation' instead of 'approach'
p.10, l.23: add 'for Eucalypt' after 'emission factors'

Thank-you for reading the paper and submitting one of the best reviews I've ever received. I would be happy to include all your suggestions to improve the understanding of the article.

Response to reviewer #2

The paper is clearly written and well structured. Methods and results are presented in a concise manner, and the conclusions drawn are sound. I did not find anything of concern while conduction my review and hence recommend for publication after the two following minor comments are addressed:

Thanks to reviewer #2 for your comments. Reviewer #2 touches on a couple of items I have thought a great deal about; choice of chemistry scheme and the boundary layer height in the model.

Detailed comments
P 2 L 26: The effect of soil moisture on plant emissions seems to be an unknown which could potentially have considerable influence on predictions. Even if the model would include it - how good is the soil moisture in the model?
There are two issues here: the soil moisture code within MEGAN and the soil moisture parameter which enters the CTM from the meteorological component (CCAM). This particular version of MEGAN v2.1 returns a value of 1 for Gamma(soil), thus the soil moisture does not influence the BVOC emission. Soil moisture within CCAM comes from CABLE, the outputs of which could be coupled to MEGAN in future, subject to aligning soil types and textures. Soil moisture at 1cm, 16cm and 44cm are used. CABLE soil moisture within the ACCESS GCM has been assessed by comparison with 19 other models in the CMIP5 evaluation and found to lie at the median of the model ensemble mean (pers comm, Ian Watterson, CSIRO). The CABLE terrestrial water cycle has been evaluated in the global soil wetness project GSWP-2 and found to compare well with evapotranspiration and runoff measurements (Zhang et al., 2013).

The following text has been added at page 4 line 14.
"…CABLE to provide information of surface roughness, soil moisture and leaf area index (LAI, based on MODIS data). The soil moisture parameter has been evaluated indirectly within the Global Soil Wetness Project, by comparing model evapotranspiration and runoff to measurements (Zhang et al., 2013). Whilst CABLE performed well, soil moisture remains a source of uncertainty."

And at page 4 line 37

"Note that soil moisture is used elsewhere in the CTM to calculate the dust emission flux, and could be coupled with MEGAN in the future".

p 4 l 18: CB05 is almost 10 years old now, and our knowledge on isoprene chemistry has improved considerably - IEPOX formation, ISOPOOH and the associated OH recycling directly impact the influence of isoprene on O3, and hence your evaluation. Can you assess how well CB05 performs compared to other mechanisms with a more updated isoprene chemistry? At least mention this potential source of error.

I have compared CB05's predecessor, CBIV to another five mechanisms, one of which was the Master Chemical Mechanism, and am aware of the differences the choice of mechanism can make to secondary species such as O3 (Emmerson and Evans, 2009). More recently Knote et al (2015) compared a couple of variants of the CB05 mechanism to other schemes containing improved isoprene oxidation schemes. The choice of mechanism resulted in differences in ozone and isoprene concentrations, particularly in biogenic regions. However, neither the Knote nor Emmerson papers went so far as to compare with measurements nor make an assessment of which scheme was 'best'. I am looking into adapting the most recent version of MOZART into the CSIRO CTM as I would like to calculate MVK and MACR separately, and consider the more recent research into isoprene oxidation pathways particularly the isoprene nitrates.

I will add the following text to page 4 line 21.
"The CB05 mechanism treats the production of a lumped isoprene oxidation product only, simplifying the chemistry. More recent schemes consider explicit oxidation products which can affect the production of ozone and nitrate species. The CB05 mechanism and its predecessor CBIV, have been compared with other schemes in Emmerson and Evans (2009) and Knote et al. (2015), but not against measurements. Choice of chemistry scheme can introduce uncertainty, which could be explored in future work".

p 8 l 19: Do you have any evaluation of the boundary layer height performance of the modeling system? The modelled concentrations are highly sensitive to this parameter, and especially its dynamic behavior (i.e. the collapse at dusk) can easily be wrong in the model.

We do not have any measurements of boundary layer height for these field campaigns. We included the ratios of isoprene and the isoprene products at figure 9 with the observations as this exercise removes the dilution effects, and still compared well.

However, we have looked at vertical potential temperature profiles from aircraft taking off from Sydney airport (AMDAR data) as a proxy to compare the model with. The modelled and observed potential temperature profiles compare reasonably well. However, the aircraft take off towards the sea and there is significant horizontal transport of the plane between the readings. The boundary layer can be inferred from these plots by eye, but we found this too subjective. We could also infer the dilution of the atmosphere by inserting a radon source to the model and comparing with radon measurements for SPS1, SPS2 and MUMBA. This is something I intend to do in future.

The following text has been added to the supplementary section, page 3 line 14.
"There are no direct measurements of boundary layer height for these field campaigns. The model boundary layer height has been compared with vertical potential temperature profiles from aircraft taking off from Sydney airport (AMDAR). From a small sample, the overall profiles compare reasonably well (not shown). However, the aircraft generally take off towards the sea and there is significant horizontal displacement of the plane between the potential temperature readings. We assess that horizontal gradients in temperature and boundary layer height in this coastal region considerably confuse the issue of resolving the

boundary layer depth at Westmead, a site 26 km inland. Thus at this stage boundary layer height verification is not possible."

Figures: Is it possible that the figures are copy-pasted from Excel or similar? Please improve their quality (spurious frames around them, resolution) to publication standards.

Agreed, I will work on improving the resolution of the images.

Response to Reviewer #3

This paper investigates the biogenic isoprene and monoterpene emissions in Southeastern Australia using the emissions model, MEGAN, and regional chemical transport models. The model output concentrations, and in one case fluxes, are compared to observations made at four field campaigns in the region. Overall, this is a valuable exercise that highlights the uncertainties in the biogenic emission estimates in Australia and highlights the lack of information needed to constrain the current models. This study is relevant to the readers of AC&P and is appropriate for publication in the journal. I recommend that this paper be published with minor edits. I provide my detailed comments here.

Thank you to reviewer #3. I am happy to include all your editorial comments and respond to your detailed questions below.

In the introduction, the MEGAN version 2.1 paper, Guenther et al. 2012, is often referenced (for example, page 1 line 32 and page 2 line4).  However, there are earlier papers that introduce the ideas discussed that should be included (i.e., earlier Guenther et al. papers from the 1990's and the MEGAN version 1 paper, Guenther et al. 2006.).

I have included the Guenther et al 2006 and Guenther et al 1995 references to page 2 line 5.

On lines 23 of page 2, Muller et al. (2008) found overestimates of isoprene. How was this determined, and with what observations?

Muller used MEGAN v2 and compared the modelled formaldehyde column to GOME satellite observations. I have rewritten the sentence at page 2 line 23 to read

"Muller et al. (2008) found an overestimate of isoprene across northern Australia by comparing MEGANv2 to GOME satellite measurements of formaldehyde, and in subsequent work estimated the magnitude of this over-prediction to be a factor of 2-3 in January (Stavrakou et al., 2015)"

The outline of the high resolution model grids would be interesting to see on Figure 1.

Done.

Page 5: Why were the Acacia species in Australia assigned the lower emission rates?

Acacia species in Australia were assigned low isoprene and monoterpene emission rates in MEGAN because the only studies we know of in the scientific literature, which have been exclusively focused on African and North American Acacias, indicate that non-negligible isoprene and monoterpene emission does occur but it is exceptional with only one high monoterpene emitter and one high isoprene emitter reported for the eight species studied. Also, Rei Rasmussen (personal

communication) has investigated isoprene emission from some common Australian Acacias and did not find any of them to be isoprene emitters.

I have altered the text on page 5 line 31 to say

"Isoprene or monoterpene emissions have not been published for any Australian Acacias but eight Acacia species from South Africa (Guenther et al., 1996; Harley et al., 2003) and the US (Guenther et al., 1999; Papiez et al., 2009) have been investigated and only one isoprene emitter and one monoterpene emitter have been identified. Based on these observations, the MEGAN model assumes low isoprene and monoterpene emission rates for Australian Acacia species."

Page 6, Section 2.3.3: The authors develop a high resolution PFT emission factor map specific to Australia based on an IGBP land cover dataset. Why was this land cover map used? It seems very old, and there are many other more recent land cover datasets available? And is this consistent with the land cover/land use datasets applied in the chemical transport models?

The Bonan et al 2002 paper was a good place to start as they showed a method to directly convert IGBP landcover into the 16 PFT classes required by the MEGAN model. It was the simplest thing to do once it became evident that the coarse resolution PFT global maps would not be suitable. I have added the following to the supplementary section, page 1 line 16:

"When emission factor maps are used, as is the case for the major biogenic species isoprene and $\alpha$- and $\beta$-pinene, the emission rates are not particularly sensitive to this PFT map. Testing the CSIRO-CTM without the emission factor maps would increase the sensitivity to PFT, which could be tested in future work. This could also be a good opportunity to test alternative land cover datasets".

Page 7, line 6: Is the broadleaf evergreen temperate tree PFT in the study dominated by Eucalypts?

Yes, Tumbarumba is surrounded by Eucalypt species of E. *delagatensis* (Alpine Ash) and E. *dalrympleana* (Mountain Gum) as described in the field campaign section on page 3 line 29.

I have altered page 7 line 13 to read "The combination of high emission factors and percentage of broadleaf evergreen temperate trees in the Tumbarumba grid (Eucalypts, section 2.1.3) enables up to 3.2 ug/m$^2$/hr of isoprene to be emitted"

Page 8: The authors perform a sensitivity test on the emissions rates. Why (or how) were the factors of 3 for isoprene and 3.5 for monoterpenes chosen? (Lines 27-30).

The factors are somewhat arbitrary, and chosen by comparing the modelled isoprene and monoterpene diurnal cycles with the observations. The increase/decrease factors varied enormously across the campaigns, however the observed monoterpene profile at Tumbarumba was ignored because it was different to the other measured monoterpene profiles. A decrease factor of 3 for isoprene suited SPS1 best whilst an increase of 3.5 suited the MUMBA monoterpenes profile best.

The text has been updated on page 8 line 33

"The factors chosen are somewhat arbitrary. A decrease factor of 3 for isoprene suited the SPS1 profile best whilst an increase of 3.5 suited the MUMBA monoterpenes profile best."

Figure 1: Which version of MEGAN emission factors are shown here?

The MEGAN emission factor maps are listed as version 2011 and dated 20 March 2013. I have added the following text to page 5 line 15 "(version 2011)"

**Editorial Comments**

I have made all the changes suggested in this section by reviewer #3

Page 2, lines 1 and 2: The sentences should read: "all of these processes"

Page 3, line 17: I suggest rewording this sentence: "Two intensive field campaigns took place: SPS1 occurred between …"

Page 4, line 30: Remove "as" before "per"

Page 5, line 35: I suggest rewording this sentence: "The PFTs listed in Table 2 of Guenther et al (2012) are comprised of various plant species that include high, moderate, low and very low emitters." I am not sure what the point is of the following sentence, and this could be removed.

We are trying to highlight how high the isoprene emission factor assigned to Australian Eucalypts (ie using approach 1 (page5, equation 1) where the model is not sensitive to PFTs) is compared to approach 2 (PFT sensitive). I have deleted the sentence and reworded to:

"…assigning Eucalypts an isoprene emission factor of 24 mg/m$^2$/hr. This is more than double the isoprene emission factor used for broadleaf evergreen temperate trees if approach 2 is used (PFT sensitive)."

Page 7, line 30: remove the comma after "dominate"

Page 8, lines 1-2: The wording of this should be changed so that the references identified are properly cited. For example: "
[revised manuscript text omitted]

[Figure]

Figure 4 Emission rates of isoprene and monoterpenes per PFT within each campaign's inner domain (180km x 180 km).

[Figure]

Figure 5 Time series of observed and modelled isoprene (left) and monoterpenes (right) for each field campaign. SPS1 = blue, SPS2 = red, MUMBA = yellow, Tumbarumba = green. Y-axis for isoprene during MUMBA restricted to 10ppb, as modelled peak is 38 ppb on 8.1.13.

[Figure]

Figure 6 Diurnal cycles of isoprene (left) and monoterpene (MT, right) emission fluxes from three days of eddy covariance measurements at Tumbarumba during November 2006. Modelled emission fluxes are plotted from the same time period.

[Figure]

Figure 7 Campaign average diurnal cycles for isoprene (left), monoterpenes (middle) and the ratio of isoprene carbon to monoterpene (MT) carbon (right). (a)=SPS1, (b)=SPS2, (c)=MUMBA (d)=Tumbarumba. F2= percentage within a factor of 2 between observations and base run.

[Figure]

| | SPS1 | | SPS2 | | MUMBA | | Tumba. | |
|---|---|---|---|---|---|---|---|---|
| | Base | EF | Base | EF | Base | EF | Base | EF |
| Isoprene | 1.2 | -0.4 | 1.1 | -0.5 | 2.5 | -0.3 | 3.9 | 0.4 |
| Isoprene Products | 2.3 | -0.1 | 1.9 | -0.4 | 2.8 | -0.04 | - | - |
| Monoterpenes | -0.5 | 0.5 | -0.5 | 0.5 | -0.6 | 0.03 | -0.2 | 1.5 |

Figure 8 Quantile-quantile plots to show relationship between modelled and observed biogenic gases. The base run are dots, the emission factor sensitivity study are the dashes. The solid black line = 1:1; dashed black lines indicate ± a factor of 2. Note: isoprene products are MVK + MACR. The y-axis on isoprene chart is reduced to 15 ppb to aid visual comparison, as modelled MUMBA data reaches 38 ppb.

[Figure]

Figure 9 Scatterplots of modelled and observed ratios between isoprene and the isoprene products, with $r^2$ correlation coefficients. EF = emission factor sensitivity test. Note, x and y axes restricted to 5 ppb and 2.5 ppb respectively.